# Human Anthrax: Update of the Diagnosis and Treatment

**DOI:** 10.3390/diagnostics13061056

**Published:** 2023-03-10

**Authors:** Mehmet Doganay, Gokcen Dinc, Ainura Kutmanova, Les Baillie

**Affiliations:** 1Department of Infectious Diseases, Faculty of Medicine, Lokman Hekim University, 06510 Ankara, Turkey; 2Department of Medical Microbiology, Faculty of Medicine, Erciyes University, 38039 Kayseri, Turkey; gokcendinc@yahoo.com.tr; 3Department of Molecular Microbiology, Genome and Stem Cell Center, Erciyes University, 38280 Kayseri, Turkey; 4Department of Infectious Diseases, International Higher School of Medicine, Bishkek 720010, Kyrgyzstan; kutmanova@yahoo.com; 5School of Pharmacy and Pharmaceutical Science, College of Biomedical and Life Sciences, Cardiff University, Cardiff CF10 3NB, UK; bailliel@cf.ac.uk

**Keywords:** anthrax, epidemiology, clinical features, diagnosis, treatment

## Abstract

Anthrax is one of the most important zoonotic diseases which primarily infects herbivores and occasionally humans. The etiological agent is *Bacillus anthracis* which is a Gram-positive, aerobic, spore-forming, nonmotile, rod-shaped bacillus. The spores are resistant to environmental conditions and remain viable for a long time in contaminated soil, which is the main reservoir for wild and domestic mammals. Infections still occur in low-income countries where they cause suffering and economic hardship. Humans are infected by contact with ill or dead animals, contaminated animal products, directly exposed to the spores in the environment or spores released as a consequence of a bioterrorist event. Three classical clinical forms of the disease, cutaneous, gastrointestinal and inhalation, are seen, all of which can potentially lead to sepsis or meningitis. A new clinical form in drug users has been described recently and named “injectional anthrax” with high mortality (>33%). The symptoms of anthrax in the early stage mimics many diseases and as a consequence it is important to confirm the diagnosis using a bacterial culture or a molecular test. With regards to treatment, human isolates are generally susceptible to most antibiotics with penicillin G and amoxicillin as the first choice, and ciprofloxacin and doxycycline serving as alternatives. A combination of one or more antibiotics is suggested in systemic anthrax. Controlling anthrax in humans depends primarily on effective control of the disease in animals. Spore vaccines are used in veterinary service, and an acellular vaccine is available for humans but its use is limited.

## 1. Introduction

Anthrax is an ancient zoonotic disease which primarily infects herbivores with humans occasionally being infected. While naturally occurring, it remains a health problem in low- and middle-income countries; its potential misuse as a biological weapon puts all communities at risk [1,2,3,4,5]. Although the human form of the disease is rarely seen in western countries, human cases have been reported. The most notable being the anthrax postal attack in the United States of America in 2001 [6], and the outbreak of injectional anthrax associated with spore-contaminated heroin in 2009–2010 [7].

The causative agent of anthrax is a bacteria called *Bacillus anthracis*, which belongs to the genus Bacillus. The organism infects herbivores through contact with contaminated soils and/or water. Humans become infected through contact with ill or dead animals and their contaminated products. *B. anthracis* is an aerobic, Gram-positive, spore-forming, non-motile, rod-shaped bacillus. The bacteria are easily grown at 37 °C on blood or nutrient agar. The organism exists in two physical forms, the biologically active vegetative form and the biologically inert spore form. It is the vegetative form which is seen in the tissue of infected individuals and is responsible for the pathology associated with the disease. As the animal succumbs to infection the vegetative form converts into inert, resistant spores which provide a lifeboat for the organism until it is able to infect a new host. The relative resistance of *B.*
*anthracis* spores to environmental conditions such as drought, heat, rain, cold, radiation and disinfectants is one of the reasons why this organism has been be explored as a potential biowarfare agent [1,8]. Figure 1 summarizes the natural life cycle of *B. anthracis*. 

The two principal virulence factors of *B. anthracis* are tripartite toxin and an antiphagocytic polypeptide capsule, the genes for which are carried on two plasmids designated as pX01 (182 kb) and pX02 (95 kb), respectively. Loss of either of these plasmids reduces the virulence of the organism. The tripartite toxin comprises protective antigen (PA), lethal factor (LF) and edema factor (EF). The role of PA is to transport LF and EF inside target cells where they interact with essential cellular pathways. The toxins are secreted the during multiplication of the vegetative *B. anthracis* and are responsible for the characteristic symptoms of anthrax [1,8,9].

The aim of this study is to review the current literature on human anthrax and to provide an update on diagnosis and current treatment options. For this purpose, PubMed, Web of Science and Google Scholar were searched using the keywords (alone and in combination) of “anthrax, epidemiology, diagnosis, treatment and therapy” since 2010. Related articles were selected for this review.

## 2. Epidemiology

Human anthrax is classified under two main headings: agricultural or industrial anthrax. In parts of the world with access to effective healthcare and veterinary services, cases of human infection are rare. Unfortunately, human infection still occurs in low- and middle-income countries where the disease is endemic in livestock and it can be found in the environment. Infection is common in farm animals (cattle, sheep, goats, horses, etc.) and wildlife (elephant, bison, buffalo, zebras, etc.) with occasional outbreaks in humans, with recent cases being reported from most areas of the world excluding northern and central Europe [1,2,5,10,11,12,13,14,15,16,17]. Industrial anthrax is the result of occupational exposure to spore-contaminated animal products such as wool and hair, which accounted for 50% or more of human cases until late in the 20th century [18] (Table 1).

## 3. Clinical Presentations

Naturally occurring anthrax is seen in one of three classical clinical forms: cutaneous, gastrointestinal and inhalational. The severity of infection depends on the innate and specific immunity of the patient, virulence and the number of infecting bacteria [1,8,9]. The majority of cases (>95%) are cutaneous in nature, with a mortality rate of less than 3–5% due to the availability of effective antibiotics [8,12,18]. As rare complications, sepsis and meningoencephalitis can develop due to spread from the primary lesion [1,7,8,10,12]. The incidence of the other forms of infection is noted as 12% for inhalation anthrax, 5% for gastrointestinal anthrax and 4% for primary meningitis. Injectional anthrax is a newly described clinical form reported in heroin users as a consequence of injecting spore-contaminated heroin with a mortality of 9–33% [7,18].

### 3.1. Cutaneous Anthrax

The incubation period is between 2–7 days (range 1–19 days) with the majority of the lesions occurring on exposed areas of the body such as the hands, arms, face and neck. A lesion begins as a pruritic papule and typically progresses to a ring of vesicles surrounded by erythema and edema within 2–4 days (Figure 2). Some lesions may be severe and extended for some distance (Figure 3). Extensive edema and toxemia can be seen in cases where the lesions occur on the face and neck (Figure 4). The formation of the eschar its subsequent resolution can take 2–6 weeks, regardless of treatment [1,8,16,19].

The differential diagnosis of cutaneous anthrax should consider staphylococcal and streptococcal skin and lymph node infections, erysipelas, orf, syphilitic chancre, cutaneous tuberculosis, ecthyma gangrenosum, ulceroglandular tularemia, plague, glanders, rickettsial infection and rat-bite fever [1,8,12,16].

### 3.2. Gastrointestinal Anthrax

The infection occurs within 3–7 days following the ingestion of *B. anthracis* within contaminated food or drinks. Lesions can be seen any point along the gastrointestinal tract. Two clinical forms are described in the literature: oropharyngeal and gastrointestinal [1,8,10,18].

The clinical features of oropharyngeal anthrax are fever, sore throat, dysphagia, hoarseness, painful regional lymphadenopathy, soft tissue edema and swelling in the neck. Streptococcal pharyngitis and tonsillitis, parapharyngeal abscess, Vincent angina, Ludwig angina, diphtheria and deep tissue infection as potential causes can be eliminated by the isolation of *B. anthracis* from the lesions [8,20,21].

The initial symptoms of intestinal anthrax include fever, nausea, vomiting, anorexia and diarrhea. As the infection progresses, symptoms include acute abdominal pain, hematemesis, bloody diarrhea and massive ascites followed by toxemia and shock which results in death. The lesions occur most commonly on the wall of the terminal ileum or cecum. The diagnosis should be confirmed by the isolation of *B. anthracis* from blood, ascites and the lesions [1,8,22,23].

### 3.3. Inhalation Anthrax

Although rare, this clinical form is mostly seen as a consequence of industrial exposure or an intentional release. The mortality is recorded at over 80% despite treatment. Following an incubation period of 1–7 days, nonspecific initial symptoms including mild fever, fatigue, malaise, myalgia, nonproductive cough and some chest or abdominal pain are seen. The disease progresses to the severe phase which is characterized by high fever, toxemia, dyspnea and cyanosis. Widening of the mediastinum is described as a typical finding of inhalation anthrax. Pleural effusion and parenchymal infiltrations can also be seen. Hypothermia and shock develop and ultimately result in death. Meningitis may develop as a complication in up to half of patients. Inhalation anthrax mimics community-acquired pneumonia and many diseases involving the pulmonary system [1,8,18].

### 3.4. Injectional Anthrax

This refers to a new clinical form of anthrax in which soft tissue is infected at the injectional site and leads to toxemia and sepsis. Gas gangrene, necrotizing soft tissue infections and severe cellulitis should be discounted [8,10,18]. 

## 4. Diagnosis

The procedures for the diagnosis of anthrax should be as follows: patient’s history, clinical examination for signs and symptoms, routine laboratory examination, radiological examinations and microbiological testing. A history of travel, residence in an endemic region, a job which involves working with animals, exposure to sick or dead animals and the handling of contaminated animal materials could indicate anthrax. Suspected cases should be confirmed by the collection of appropriate samples from the patient’s lesions and subsequent laboratory examination according to the WHO Guidance [1,8,10,16,18]. These samples include swabs from cutaneous lesions, blood, sputum, pulmonary effusion or bronchial biopsy specimens in cases of suspected inhalational anthrax. They also include samples from oropharyngeal lesions, ascites fluid, feces and vomit in suspected cases of intestinal anthrax, and cerebrospinal fluid when meningitis is suspected [1,8].

When biochemical and blood parameters are evaluated, the leukocyte count is usually less than 10 × 10^3^ cells/µL in mild cutaneous cases. In complicated cutaneous infections, toxemic shock, systemic anthrax; leukocytosis with neutrophilia, hypoalbuminemia, hyponatremia, and rising aspartate aminotransferase (AST) and alanine aminotransferase (ALT) levels may be detected. If severe sepsis develops, leukopenia, thrombocytopenia and disseminated intravascular coagulation (DIC) may occur [8,24].

The identification of the pathogen is based on a combination of microscopy and culture. The vegetative form of the bacteria appears as a Gram-positive rod-shaped organism. Confirmation of the presence of a capsule which surrounds virulent forms of the bacterium can be confirmed using polychrome methylene blue or Indian ink. Staining with methylene blue reveals the presence of blue-black, square-tipped bacilli surrounded by a pink capsule. The organism can grow on a range of general culture media including blood agar. The BACTEC™ FX40 device, an automated blood culture apparatus, is widely used in microbiology laboratories to recover the pathogen from clinical samples [8,24,25].

For specimens in which the bacteria are likely to present in the company of other micro-organisms, a selective agar is recommended such as Polymyxin-Lysozyme-EDTA-Thallous acetate (PLET) agar. It is based on heart infusion agar supplemented with polymyxin B, lysozyme, ethylenediaminetetraacetic acid (EDTA) and thallus acetate. On this medium the bacterium produces colonies which are white, dome-shaped, round and small. Anthrax Blood Agar (ABA) containing cycloheximide, polymyxin B, trimethoprim and sulfamethoxazole is another selective option. When cultured on this medium, the organism produces white or gray non-hemolytic colonies. R&F Anthrax Chromogenic Agar (ChrA) containing cycloheximide, polymyxin B and X-indoxyl-choline phosphate (X-CP) is another option. The role of X-CP in the medium is to detect the presence the phosphatidylcholine phospholipase C enzyme, which is secreted by *B. anthracis*, *B. cereus* and *B. thuringiensis*. Following 24 h incubation, colonies of *B. anthracis* appear frosted glass-like in appearance and are cream to faded blue in color, with developing white edges following a further 24 h incubation. Finally, Chromogenic *Bacillus Cereus* Agar and Cereus Ident Agar can be used as selective media. These media include a chromogenic substrate, 5-bromo-4-chloro-3-indolyl-ß-glucopyranoside, which is degraded by the ß-glucosidase enzyme which is expressed by most species of Bacillus. *B. anthracis* colonies are a white-creamy color on these media [8,24,26].

*B. anthracis* can be differentiated from other *Bacillus* species using a range of simple first-line laboratory tests which include gamma phage susceptibility, catalase production, lack of motility, lack of hemolytic activity when cultured on blood agar and susceptibility to penicillin [24,27]. Although microbiological techniques are the best way to identify the bacteria, they occasional generate ambiguous results, particularly when attempting to differentiate the pathogen from closely related strains of *Bacillus cereus*. Many phenotypic features of *B. anthracis* may also be displayed by some *B. cereus* strains. For example, there have also been reports of strains of *B. anthracis* that are hemolytic and are resistant to both penicillin and the gamma phage.

To determine the sensitivity of the organism to antibiotics and determine the most appropriate treatment regimen it is necessary to cultivate the agent. The EUCAST (The European Committee on Antimicrobial Susceptibility Testing) has issued guidelines for the rapid antimicrobial susceptibility testing (RAST) of bacteria such as *B. anthracis*, where rapid results are required to reduce morbidity and mortality [25,28]. Shifman et al. (2021) performed antimicrobial tests (disc diffusion and E-test) directly from blood cultures containing *B. anthracis* and confirmed the feasibility of this approach [25].

The use of DNA amplification-based PCR (Polymerase Chain Reaction) and Real-time PCR tests can be used for the definitive and rapid diagnosis of *B. anthracis* in clinical and environmental specimens. Diagnostic targets include specific DNA regions in the following genes: *pagA* (pXO1), *cap*B (pXO2), *cap*C (pXO2) and *Ba*813 (chromosomal) [29,30,31,32]. More recently it has been reported that newer isothermal DNA amplification techniques such as RPA (recombinase polymerase amplification), HDA (helicase-dependent amplification) and LAMP (loop-mediated isothermal amplification) can be used. Rapid DNA-based methods are particularly useful for the confirmation of the cause of infection in patients who have been treated with antibiotics which would prevent the bacteria from growing on culture. In addition to DNA-based methods, immunological approaches can also be employed to diagnose the presence of the pathogen. These include flow cytometry analysis using fluorescently labeled antibodies, FRET (Förster resonance energy transfer), ELISA (Enzyme-linked immunosorbent assay), Luminex test, MPFIA (magnetic particle fluorogenic immunoassay) and ABICAP (Antibody Immuno Column for Analytical Processes) immunofiltration. Antibody-based lateral flow devices have been developed for the screening of environmental samples but are not suitable for use with clinical samples [33,34,35].

In recent years there has been a focus on the development of biosensors capable of the rapid and specific detection of *B. anthracis*. There are currently four types of biosensor platforms: Genosensors (Nucleic Acid Probes), Immunosensors (Antibody Probes), Aptasensors (Aptamers) and Peptide-Nucleic Acid Chimera Probes (PNAs). They employ a range of signal generation approaches which include electrochemical (amperometric, potentiometric and conductometric), optical and piezoelectric. Genosensors work on the principle that the binding of pathogen-specific DNA to a probe generates a signal. To date, genosensors have been developed which target *pagA*, *lef* and BA813 [32,36,37,38]. Impedimetric aptasensors have also been used for single-step identification of *B. anthracis* spore simulants [39]. Antibody probes that recognize *B. anthracis*-specific antigenic structure have been incorporated into immunosensors such as an ultrasensitive portable capillary biosensor (UPAC) and a magnesium niobate-lead titanate/tin (PMN-PT/Sn) piezoelectric microcantilever sensor (PEMS) [40,41]. Matrix-assisted laser desorption ionization time-of-flight (MALDI-TOF) mass spectrometry (MS) has also been used to detect the pathogen in clinical and environmental specimens [42].

While the detection of an antibody response to the pathogens toxins has little diagnostic value in the early stages of the disease, serum samples should be obtained at 0 to 7 days of illness and at 14 to 28 days to allow the clinical team to confirm diagnosis. There are several rapid anthrax-PA ELISA kits which have been approved by the US FDA for this purpose [24].

## 5. Management

Anthrax, appearing under the names of “sacred fire” (ignissacer) or “Persian fire” (ignispersicus), has been known since antiquity, and was described in the Murrain of Noricum by the Roman poet Virgil. In 1788, S.S. Andrievsky discovered that anthrax in humans and animals had the same etiology, and F. Brawell proved the possibility of anthrax transmission from animals to humans. A pure culture of the pathogen was obtained by Koch in 1876, and in 1881 Pasteur developed a live vaccine against the disease in animals [43]. In spite of these grand discoveries, the issues of an effective anthrax treatment in humans had yet to be addressed. In the early 1900s, the development of therapeutic sera was one of the most significant practical objectives in various laboratories all over the world. At that time, along with Sclavo’s serum, surgical excision and antiseptic chemicals were concurrently used in the practice. Salvarsan and neoarsphenamine in a combination with Sclavo’s serum were used for anthrax treatment up to the 1940s, when eventually sulfonamides and antibiotics replaced them. Nowadays, antibiotics are considered to be the main therapy for patients with anthrax [44].

Currently, antibiotic therapy is still considered the main option for anthrax treatment. However, the approach to anthrax treatment differs from other bacterial infections due to such features as toxin production, the antibiotic resistance problem and high frequency of meningitis occurrence. In vitro, *B. anthracis* clinical isolates are susceptible to various antibiotics including penicillin, aminoglycosides, macrolides, quinolones, carbapenems, tetracyclines, vancomycin, clindamycin, rifampicin, cefazolin and linezolid. The recommendations for antibiotic prescription differ depending on the site and severity of the disease [16,45,46]. Thus, in naturally occurring anthrax cases, penicillin G and amoxicillin are the first-choice drugs while doxycycline and ciprofloxacin are the alternative agents, in a treatment scheme with a duration of 5–7 days for mild and uncomplicated cutaneous anthrax cases, and 10–14 days for complicated cutaneous and systemic anthrax cases [12,47,48,49]. In mild and uncomplicated cases with cutaneous anthrax, an oral antibiotic is suggested. In severe cases such as inhalational or gastrointestinal anthrax, meningoencephalitis, sepsis or cutaneous anthrax with extensive edema, antibiotics should be administered intravenously; when the fever has subsided to normal, antibiotic therapy may be switched to oral. In severe cases or internal organ anthrax, initial antibiotic choice should be combined with one or two of the following antibiotics: penicillin, ampicillin, ciprofloxacin, imipenem, meropenem, vancomycin, rifampin (rifampicin), clindamycin, linezolid or aminoglycoside [1,8,10,12].

The recommended antibiotic regimens in adult cases are as follows: procaine penicillin G, 0.6–1.2 M units intramuscularly every 12–24 h; amoxicillin 500 mg orally every 6–8 h; doxycycline 100 mg intravenously or orally every 12 h; ciprofloxacin 200–400 mg intravenously every 12 h, followed by 500–750 mg orally every 12 h [48,49,50]. If such regimens are followed, cutaneous lesions usually become sterile within the first 24 h with edema regressing within 24 to 48 h. However, it should be noted that although due to early treatment the size of the lesion will be limited, the evolutionary stages of the lesion will not be altered. Penicillin G is considered to be the first-choice antibiotic in these cases and should be prescribed at the dosage of 2400 mg (4 M units) every 4–6 h by infusion until the resolution of patient’s symptoms and normalization of the temperature [1,12,48].

Ciprofloxacin and doxycycline are considered to be the first-choice agents in cases of biological weapon or bio-terrorism-related anthrax, and are prescribed in the following regimens: ciprofloxacin 200–400 mg intravenously every 12 h, followed by 500–750 mg orally every 12 h; doxycycline 100 mg intravenously or orally every 12 h with the treatment duration of 42–60 days [47].

### 5.1. Cutaneous Anthrax

Antibiotic administration is the main treatment for naturally occurring anthrax. In mild or uncomplicated cases, monotherapy with intramuscular penicillin G or the oral antibiotics doxycycline or ciprofloxacin are effective. In severe and complicated cases antibiotics should be administered intravenously. During the acute inflammatory period cutaneous lesions should be dressed and covered with a sterile wrap; surgical intervention should be avoided as it can lead to dissemination and a poor outcome [1,50]. However, the surgical debridement of infected soft tissue in combination with antibiotics and supportive therapy may be life-saving in cases of injectional anthrax [7,8,10,18].

In cases where the head and neck are affected, adjunctive corticosteroids may be prescribed as an anti-edema treatment to avoid possible serious complications. Furthermore, careful monitoring for airway compromise should be carried out as respiratory support for airway edema may be required [1,8,51].

### 5.2. Gastrointestinal Anthrax

As clinical manifestations of gastrointestinal anthrax vary largely, imitating a range of different diseases, physicians of the endemic areas should be especially aware of this clinical form. Due to the lack of awareness, patients undergo unnecessary interventions such as abdominal surgery because of acute abdomen syndrome or, on the contrary, fail to receive adequate medical treatment if only mild diarrhea is present. In case of gastrointestinal anthrax, initial antibiotic therapy is suggested, using a combination of penicillin G with streptomycin or other aminoglycosides. To improve the outcomes of patients diagnosed with gastrointestinal anthrax, the treatment should be followed as [1]: a. Give an appropriate antibiotic in adequate dose intravenously. b. Careful monitore fluid, electrolyte and protein losses and in-time replacement for patient. c. To consider early surgical resection of the necrotic intestine. Surgical treatment in the form of wide resection of the infected and necrotic parts of the intestine with primary anastomosis may be lifesaving.

### 5.3. Inhalation Anthrax

In case of inhalation anthrax, penicillin G may be used in the combination with clindamycin or clarithromycin initially. In case of allergy to penicillin, other alternative regimens can be used in the therapy.

In inhalation anthrax, patients may require mechanical ventilation due to respiratory failure caused by reaccumulating pleural effusions. However, the necessity and duration of ventilation may be reduced by performing pleural space drainage [51]. According to Holty, J.E. et al., this procedure improved outcomes in a series of cases by decreasing the level of lethal factors and mechanical lung compression [52,53]. Thus, performing an aggressive pleural fluid drainage at the earliest possible time after the detection of any clinically or radiographically apparent pleural effusions is highly recommended. It should be noted that chest tube drainage is more preferred compared to thoracentesis as prolonged drainage may be required. In order to remove gelatinous or loculated collections, thoracotomy or video-assisted thoracic surgery is recommended [51].

### 5.4. Anthrax Meningoencephalitis

A combination of antibiotics is preferred with one of them being able to penetrate to the central nervous system. If anthrax meningoencephalitis is suspected, empiric treatment should include three or more antimicrobial drugs showing an adequate level of CNS penetration with one of them having bactericidal activity, and another one should be a protein synthesis inhibitor. The duration of treatment for anthrax meningoencephalitis should be carried out for at least two weeks or until the clinical stability of the patient’s condition. Considering the high frequency of fatal outcomes, some experts recommend to provide at least a three-week treatment in cases when the diagnosis of meningitis cannot be excluded. An algorithm is suggested by the experts for the diagnosis of meningitis [51]. Due to the good CNS penetration and relatively low level of antimicrobial resistance, ciprofloxacin (400 mg every 8 h) is recommended as a component with bactericidal activity in the treatment scheme of anthrax meningoencephalitis, while levofloxacin (750 mg every 24 h) and moxifloxacin (400 mg every 24 h) are alternatives [51,54]. A combination of penicillin G or a fluoroquinolone with rifampicin is currently considered the first-choice treatment in such cases due to its effective activity against *B. anthracis* and fast penetration into the cerebrospinal fluid. The recommended dose of crystalline penicillin G is 20–24 million units per day divided for intravenous administration every 2–4 h, and rifampicin should be prescribed in a dose of 600–1200 mg per day (can be administered intravenously or via an enteral tube if needed) [1,12]. Due to the good CNS penetration and relatively low level of antimicrobial resistance, ciprofloxacin (400 mg every 8 h) is recommended as it has bactericidal activity, with levofloxacin (750 mg every 24 h) and moxifloxacin (400 mg every 24 h) as alternatives if required [51,54].

Supportive therapy plays an important role in anthrax meningoencephalitis as respiratory support and anti-edema therapy for the brain may be life-saving measures in most cases. In order to provide supportive therapy, such measures as early initiated assisted respiration, maintaining of fluid and electrolyte balance and anti-edema treatment (100 mL of 20% mannitol intravenously every 8 h and 100 mg of hydrocortisone every 6 h) are essential [1].

In addition to antibiotic therapy, specific antitoxin serum for anthrax may be used in injectional anthrax and systemic anthrax. A polyclonal anthrax immune globulin intravenous (AIGIV-Anthrivig) drug and two monoclonal antitoxins, raxibacumab and obiltoxaximab, were developed in the USA and have shown beneficial effects in an animal model [55].

Vaccines to project humans against anthrax are available. Attenuated live spore vaccines are available in China and Russia to protect at-risk individuals. Due to concerns over the reactogenicity of spore-based vaccines in humans, acellular vaccines have been developed in the UK and the USA. Protective antigen is the principal protective immunogen in these licensed human vaccines [56,57,58,59]. Both require multiple doses to induce protection, and because of the manner by which they were developed, they are relatively crude products containing trace amounts of LF, EF and other bacterial antigens [60]. While primarily intended to provide protection against infection, they have a role to play in the treatment of individuals who have inhaled spores of the pathogen. The US CDC recommends a post-exposure regimen of 60 days of appropriate antimicrobial prophylaxis combined with 3 subcutaneous doses of the anthrax vaccine [61]. The reasons for this are because the spore form of the pathogen, which is resistant to antibiotics, could remain dormant in the lungs until antibiotic treatment has stopped, at which time the spore could germinate to initiate a new infection. The administration of the vaccine allows the individual to develop a protective immune response which is able to deal with any germinating bacteria.

## 6. Conclusions

Anthrax is still an endemic disease in middle- and low-income agricultural countries. The main transmission of anthrax to human is by contact with ill or dead animals and contaminated animal products. The majority of clinical forms is cutaneous anthrax. Suspected cases should be confirmed by laboratory tests. Human isolates are generally in vitro susceptible to many antibiotics. Penicillin G, amoxicillin, ciprofloxacin and doxycycline are widely using in the clinical practice of naturally occurring anthrax. The prevention of human anthrax is based on the control of animal infection, education of animal owners and occupational risk groups. For example, ill or dead animals ought not be slaughtered, skinned or butchered for consumption of their meats or have their products used. The control of infections in animals consists of close surveillance, vaccination of animals against anthrax and good veterinary practices, which include the burying or cremation of infected animal carcasses and the use of effective decontamination and disinfection procedures. For human immunization, an acellular vaccine is only available in the UK and the USA, and it is used mostly in occupational and military settings. A live spore vaccine for immunization of human has been produced and used in China and Russia.

## Figures and Tables

**Figure 1 diagnostics-13-01056-f001:**
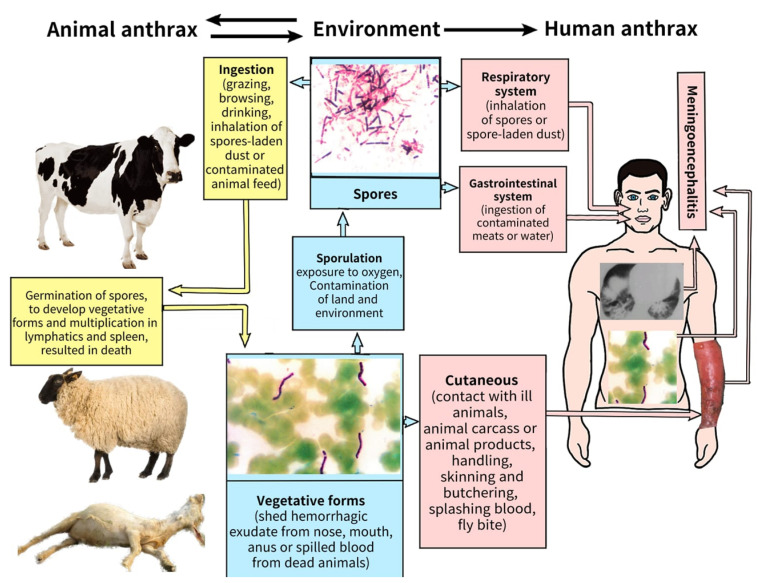
The life cycle of *Bacillus anthracis* in nature. Soil is the main reservoir of the pathogen and is contaminated by spores released from the carcasses of infected animals. Animals grazing on spore-contaminated land become infected resulting in a new cycle of infection, death and release of spores which can potentially contaminate a new location. Wild carnivores and scavenger birds and flies may also contribute to the spread of spores. Humans can be infected by contact with infected animals or contaminated animal products. The figure was created by Fatma Beyzanur Koyuncu, Medical student in Lokman Hekim University, Ankara).

**Figure 2 diagnostics-13-01056-f002:**
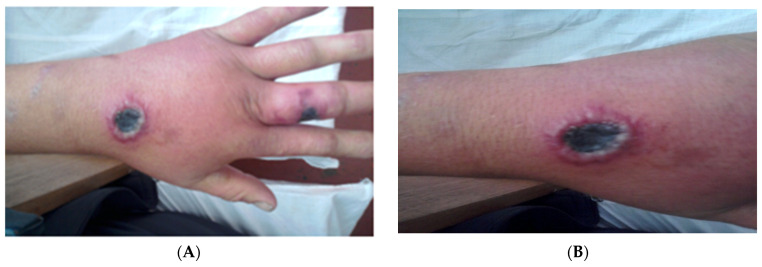
Typical lesions of cutaneous anthrax on the hand (**A**) and leg (anterior side) (**B**). The lesions are characterized by a central eschar with a ring of vesicles and surrounding edema that is characteristically painless. (Images supplied by Professor Ainura Kutmanova, and Dr. Saparbai Zholdoshev).

**Figure 3 diagnostics-13-01056-f003:**
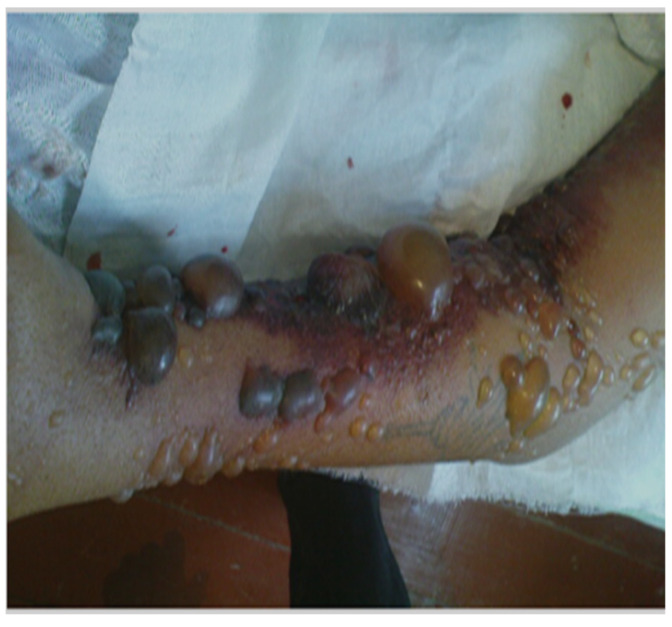
Typical appearance of a severe form of cutaneous anthrax lesions on the arm. Extensive erythema and oedema as well as hemorrhagic bullae can be seen (Image supplied by Professor Ainura Kutmanova and Dr. Saparbai Zholdoshev).

**Figure 4 diagnostics-13-01056-f004:**
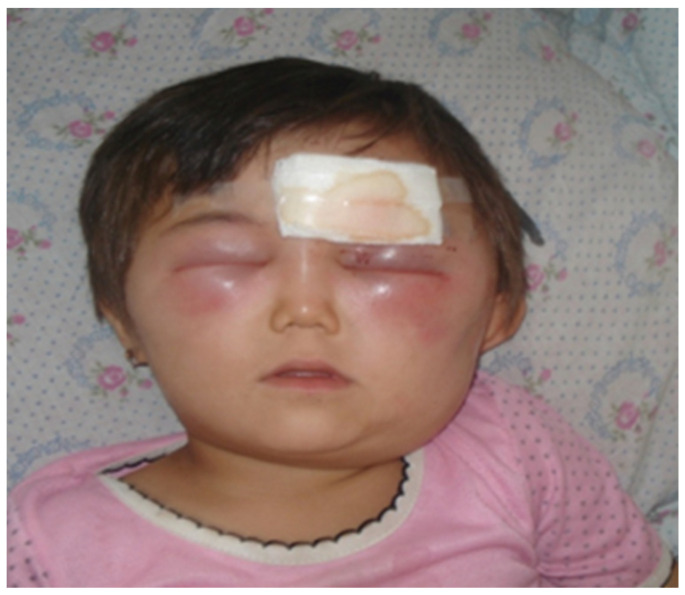
A severe form of cutaneous anthrax on the face. Extensive erythema and oedema can be seen (Image supplied by Professor Ainura Kutmanova and Dr. Saparbai Zholdoshev).

**Table 1 diagnostics-13-01056-t001:** Human anthrax: Common routes of transmission, risk factors and clinical presentations.

TRANSMISSION OF THE INFECTION	COMMENTS
Contact with dying or dead animals	Slaughtering and skinning, handling and processing of dead animals. The disposal of contaminated carcasses
Contact with contaminated animal products	Wool coats, shaving brushes, leather (e.g., drumheads made of animal skin) and bone meal (e.g., fertilizer).
Ingestion of contaminated meat	Eating contaminated, uncooked, improperly cooked meat, traditional raw food or meat products.
Self-injection	Injection of contaminated, illegal heroin
Nosocomial transmission	Human to human spread is rare
**RISK FACTORS**	
Living in an endemic area	
Agricultural occupations	Herdsman, butchers, skinners, slaughterhouse workers, diary workers, veterinarians
Industrial occupations	Tanners, leather gift makers, furriers, shoemakers, drum makers, carpet weavers, wool spinners, bone meal processers, wool textile factory workers
Illegal drug use	Injection of contaminated materials
**CLINICAL PRESENTATIONS**	**COMPLICATIONS ***
Cutaneous anthrax	Sepsis Meningoencephalitis Pneumonia
Gastrointestinal Anthrax Oropharyngeal Intestinal	
Inhalation anthrax	

* Sepsis, meningitis or pneumonia may occur due to the pathogen spreading from the primary site of infection.

## Data Availability

Not applicable.

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
