# Peer review of "Human Anthrax: Update of the Diagnosis and Treatment"

_diagnostics, 2023, doi:10.3390/diagnostics13061056_

Round 1

Reviewer 1 Report

Human anthrax: update of the diagnosis and treatment

Major comments

As title suggest you should more focus on diagnosis and treatment part.

Provide numbering to the sections and subsections.

The article should be more comprehensive with better tables. Figures are of poor quality.

Figure 1 - It should be in circular form as it it cycle...

Minor comments 

L- 34 Reform the sentence "Humans are infected with incidentally"

L- 73 Remove "(" ,before protective antigen

L-75 Remove "the", before during 

L-78 "is" instead of "was"

L- 98 Remove space

L-128 Add "," before and

L- 158 Add "," before and. Check it throughout the document.

L - 173 Remove space

L- 206 Full form of AST and ALT?

L- 208 Add "." at the end of sentence after ref.

L- 237 PC-PLC

L- 254 full form of EUCAST?

L-261 full form of PCR? Ensure that all the full forms are mentioned thorough out the document

Author Response

RESPONSE TO REVIEWERS

 On behalf of the authors of the manuscript entitledHuman anthrax: update of the diagnosis and treatment”, I would like to thank to all of the reviewers for their valuable suggestions. The manuscript revised according to the reviewer suggestions. The English text is also extensively revised by the author who has a native speaker.

REVÄ°EWER 1

Comments and Suggestions for Authors

Human anthrax: update of the diagnosis and treatment

Question: Major comments

Q1. As title suggest you should more focus on diagnosis and treatment part.

Reply. The manuscript is mainly focused on diagnosis and treatment. Some part of the text was deleted.                

Q 2. Provide numbering to the sections and subsections.

Reply. Yes, Numbered the section and subsections.

Q 3. The article should be more comprehensive with better tables.

Reply. One table was added

Q.4. Figures are of poor quality.

Reply. Figure 1 quality was a little bit improved. Unfortunately, not improver others.

Q5.  Figure 1 - It should be in circular form as it it cycle...;

Reply: We could made only little changes as seen the new form in the text.

Minor comments : All minor comments were corrected.

L- 34 Reform the sentence "Humans are infected with incidentally"

L- 73 Remove "(" ,before protective antigen

L-75 Remove "the", before during 

L-78 "is" instead of "was"

L- 98 Remove space

L-128 Add "," before and

L- 158 Add "," before and. Check it throughout the document.

L - 173 Remove space

L- 206 Full form of AST and ALT?

L- 208 Add "." at the end of sentence after ref.

L- 237 PC-PLC

L- 254 full form of EUCAST?

L-261 full form of PCR? Ensure that all the full forms are mentioned thorough out the document

Reviewer 2 Report

As this is a review article, please include more details on the vaccine(s) available for anthrax and also other FDA-approved emergency treatments like monoclonal antibody therapy (Anthim). This will make the review more comprehensive.

Author Response

RESPONSE TO REVIEWERS

On behalf of the authors of the manuscript entitledHuman anthrax: update of the diagnosis and treatment”, I would like to thank to all of the reviewers for their valuable suggestions. The manuscript revised according to the reviewer suggestions. The English text was also extensively revised by the author who has a native speaker.

 REVÄ°EWER 2.

Comments and Suggestions for Authors

As this is a review article, please include more details on the vaccine(s) available for anthrax and also other FDA-approved emergency treatments like monoclonal antibody therapy (Anthim). This will make the review more comprehensive.

Reply. Thanks to the reviewer suggestions.  The authors have added a paragraph for vaccine at the end of the management section.

Round 2

Reviewer 1 Report

The manuscript is revised substantially